# Role of Clay Mineralogy in the Stabilization of Soil Organic Carbon in Olive Groves under Contrasted Soil Management

Julio Calero [1,*], Roberto García-Ruiz [2], Milagros Torrús-Castillo [1], José L. Vicente-Vicente [3] and Juan M. Martín-García [4]

1   Center for Advanced Studies in Earth Science, Energy and Environment, University of Jaén, Campus Universitario de Las Lagunillas s/n, Edificio A-3, 23071 Jaén, Spain
2   University Institute of Research in Olive Grove and Olive Oils, University of Jaén, 23071 Jaén, Spain
3   Leibniz-Centre for Agricultural Landscape Research (ZALF), Eberswalder Str. 84, 15374 Müncheberg, Germany
4   Department of Edaphology and Agricultural Chemistry, University of Granada, Campus Cartuja s/n, 18071 Granada, Spain
*   Correspondence: jcalero@ujaen.es; Tel.: +34-953212830

**Abstract:** Cropland soils are key systems in global carbon budgets due to their high carbon-sequestration potential. It is widely accepted that clays are one of the soil components that have a significant effect on the stabilisation of soil organic carbon (SOC), owing to its surface interactions with organic molecules. However, the identification of the direct effects of clays on SOC stabilization is complicated, mainly due to the difficulty of accurately characterizing the mineralogy of clays, especially phyllosilicates. In this study, the relationships between soil phyllosilicates and functional SOC pools in woodlands and comparable olive groves, under two contrasting management systems (bare soils versus soil under cover crops) and parent materials (calcareous and siliceous), were explored. The total mineralogy of soil and clay fractions and the soil-clay assemblages were analysed through the decomposition of X-ray diffraction patterns, and were then related to four SOC pools. Total and unprotected SOC was higher in olive groves under cover crops, and this was true independent of the parent material, proving the importance of herbaceous covers in SOC sequestration in woody crops. Some significant correlations between clay minerals and SOC fractions were found. Interestingly, mixed-layer content was correlated with the biochemically protected SOC fraction ($r = 0.810$, $p < 0.05$), and this was so even when the partial correlation coefficient was calculated ($r = 0.761$, $p < 0.05$). According to the partial correlation networks (PCN), four separated clusters of variables were obtained, which joined into only one at $fdr < 0.25$. The PCNs supported the direct correlation between mixed-layer content, especially those rich in smectite, and the biochemically protected SOC fraction, suggesting that smectite layers may stabilize organic molecules. Since potassium enrichment is higher in the rooted layers of woodland and soils under cover crops, and this increase is related to the collapse of swelling layers, these soils were poorer in smectite phases than the bare soils. This also would explain why the biochemically protected SOC was more abundant in the latter.

**Keywords:** soil organic matter; cover crops; C sequestration; Mediterranean soils; SOC fractionation; XRD



## 1. Introduction

The stability of soil organic carbon (SOC) in croplands has become an important research topic after the Paris Agreement [1] due to its role in the regulation of the global carbon cycle. Despite the fact that cropland only occupies about 13% of the global mainland [2], it is one of the anthropogenic biomes with the highest carbon sequestration potential (between 0.9 and 1.85 Pg C year$^{-1}$). This could account for up to 20% of fossil fuel emissions per year [3]. The "4 per mille" initiative provides a conceptual framework to boost the organic carbon accumulation in cropland soils. Although this initiative seems theoretically

feasible, it raises some difficulties and uncertainties [4]. Indeed, not all cropland soils would achieve the goals set by the "4 per mille" initiative, as soils have a limited capability to sequester carbon, according to the relatively well-established C-saturation hypothesis [5]. The SOC saturation framework is based on several stabilization mechanisms that protect SOC against heterotrophic respiration, resulting in an increase in the mean SOC residence time. These mechanisms are driven at different scales by intrinsic (mainly texture and mineralogy) or extrinsic (climate, topography, etc.) soil properties, and should be accurately assessed for a useful prediction of soil carbon budgets [6].

Soil organic matter (SOM) stabilization depends on the molecular associations at the level of organo-mineral complexes [7]. Indeed, SOM is considered to be a continuum of organic molecules associated with minerals. This theory challenges the validity of the traditional humus-fractioning by means of chemical procedures [8]. During the last two decades, new methods—generically called 'functional fractioning methods'—have been raised [5,9], mainly based on assessing SOC dynamics, its resistance to degradation in the soil and the necessary balance between C stocks and flows in the ecosystem [10]. The empirical testing of these methods has established more accurate relationships between conceptual SOC pools and measurable SOC fractions, highlighting the critical importance of texture and clay mineralogy in the SOC turnover rate [7].

It is generally accepted that the higher the content of fine particles in the soil, the greater the stabilization of SOC [5]. However, Barré et al. [11] proved, in an extensive review, the influence of natural soil–clay minerals (mainly phyllosilicates and iron oxides) on the soil's ability to stabilize SOC, beyond its fine-grained size. Two of the three basic stabilization mechanisms are based on the physical inaccessibility and the adsorption of mineral surfaces, which seem to be conditioned by the clay composition [12]. Clay-mineral surfaces bind and chemically stabilize dissolved or molecular forms of SOC [13]. The direct linkage of SOC with soil clay layers, together with surface processes which control the formation of organo-mineral complexes, are involved in the aggregation of structural units (clay domains, clusters, micro- and macroaggregates). These units are involved in the persistence of the particulate SOC persistence through physical stabilization mechanisms, such as occlusion [5,14]. In recent studies, Plaza et al. [15] and Calero et al. [16] incubated two different Mediterranean soils, rich in smectite and illite, with a commercial humic acid. These authors found evidence that both the flocculation of clays and the adsorption of organic carbon depended on the same electrokinetic and thermodynamic processes developed on mineral surfaces, which were strongly conditioned by clay mineralogy. Therefore, a comprehensive understanding of SOC stability should consider both the physical and chemical protection mechanisms which define the different functional pools.

Studies aiming to assess the role of phyllosilicate in SOC stabilization do not usually consider in great detail the composition and relative amount of the <2 μm clay assemblages, despite the availability of suitable techniques (peak decomposition from X-ray patterns) [17]. This is mainly due to the high time-cost and intensive labour required for a full characterization of clay assemblages and their properties, such as specific surface area and cation exchange capacity, which restricts the number of samples that can be reasonably considered [6]. On the other hand, a small number of samples reduces the power of statistical tests, because many soil properties co-vary with mineralogy (and also with SOC pools); consequently, many spurious relationships might be expected [11]. To solve these restrictions, partial correlations of different orders have been applied. However, with an increasing number of variables yielded by the high-resolution DRX and the functional-fractions analyses, the complexity of partial correlation approaches based in low-order partial coefficients may become untreatable. One of the most recent and interesting applications in revealing direct correlations in complex systems (e.g., genomic) are the partial correlation networks (PCN), which can address high dimensionality systems with a low number of samples [18]. Soil biogeochemical processes possess complicated patterns of reciprocal influence between inorganic and organic soil compounds and properties;

interpretation of these could be improved by means of PCN. Nevertheless, this method has been only recently applied in soil science [19].

Olive groves are a key crop in the Mediterranean basin, where more than 70% of olive groves and world olive oil production is concentrated in just six countries: Spain, Italy, Greece, Morocco, Tunisia and Turkey. About 8 million hectares are devoted to this crop (about 3% of the total area of these countries), so changes in management practices have important environmental consequences (water withdrawal, soil erosion, carbon emission, biodiversity loss, etc.). Conventional tillage, where soils are usually left bare by mechanical and/or chemical ploughing, still predominates, favouring high rates of soil loss and low levels of soil quality [20]. For instance, in the soils in Jaen province (Southern Spain), which is the largest olive oil production area worldwide accounting for 85% of the agrarian-utile land, the organic carbon content in the topsoil is estimated to be less than 1% [21]. Regional, national and EU agrarian policies boost the implementation of management practices that might contribute to increasing the SOC levels. Among the practices for woody crops, there is political recommendation for the utilization of cover crops in the inter-canopy area of these orchards [22], which might result in a significant increase in SOC and changes in the soil–clay assemblies, significantly affecting SOC stabilization.

In this study, we explore direct, non-spurious associations between clay–mineral assemblages and functional SOC fractions in olive groves with and without cover crops, and in undisturbed woodlands as a reference, and in two contrasted parent materials, with the aim of a better understanding of the SOC stabilization processes occurring in olive grove soils.

## 2. Material and Methods

### 2.1. Sites and Samples

Olive groves at two sites, with contrasted parent material (calcareous vs. siliceous), were selected in Jaén (calcareous site) and Córdoba (siliceous site) provinces. At each site, we identified and selected: one olive grove with a long continuous history (>20 years) of bare soils (B); an olive grove that was comparable (similar landscape features, slope, climate and tree density) and nearby (<50 m), but with temporary spontaneous cover crop (C); and a semi-natural woodland (W). Orchard and woodland sites were in steep landscapes, with slope gradients ranging between 20% and 30%. The climate is Mediterranean, with hot and dry summers; the mean annual temperature oscillates between 17.3 °C and 15.1 °C, and the annual rainfall between 490 and 652 mm. Natural vegetation in the woodland is composed mainly by holm oaks (*Quercus rotundifolia*) that appear in small and dispersed patches among the orchards. Groves under bare soils were tilled, drip-irrigated, subjected to three to five chisel passes a year to a depth of 15–20 cm from early spring to autumn, and weed control was performed with residual herbicides. Olive groves with cover crops were characterized by non-tilled soils with a long-term (minimum of 15 years) use of plant cover, which was controlled only by mowing from early to late spring, and without the addition of chemical fertilizer or pesticides. The olive groves over siliceous parent material are located in the municipality of Obejo, Córdoba, belonging to the Pedroches Batolite, a granitoid complex of post-kinematic origin with granodiorite (G) lithology. The calcareous olive groves are located in the municipality of Bedmar, Jaén, over carbonate rocks of Mesozoic age in the Betic mountain range. In contrast to the homogeneous granodiorite, carbonate parent materials showed very different properties, and cropland and woodland soils were sampled separately. Soils from limestone colluvia (L) and in situ marls (M) were sampled at the calcareous sites, except for the case of a woodland calcareous site, because the significant anthropogenic pressures made it impossible to find a well-preserved natural plot over marls.

In total, eight experimental plots were evaluated (Table 1). Five composite topsoil (0–5 cm) samples were obtained in the inter-canopy area of each plot, and each sample consisted of four subsamples randomly taken within a 5 m radius.

**Table 1.** Some characteristics of the sampled plots. W stands for forest; B and C for olive groves with bare soil or with spontaneous cover crops, respectively. G, L and M stand for siliceous and calcareous limestones or calcareous marl, respectively. Masl stands for meters above sea level.

| Sample | Location | Altitude (Masl) | Parent Material | Slope (%) | Management (Vegetation) |
|--------|----------|-----------------|-----------------|-----------|--------------------------|
| WG | 38°11′37.65″/4°50′31.63″ | 711 | granodiorite | 23 | natural (holm oak woodland) |
| BG | 38°11′35.49″/4°50′33.21″ | 690 | granodiorite | 26 | conventional orchard (bare soil) |
| CG | 38°11′34.28″/4°50′29.72″ | 690 | granodiorite | 27 | organic orchard (cover crop) |
| WL | 37°46′36.42″/3°20′22.20″ | 803 | limestone colluvia | 32 | natural (holm oak woodland) |
| CL | 37°46′47.36″/3°20′31.22″ | 705 | limestone colluvia | 29 | organic orchard (cover crop) |
| CM | 37°46′51.66″/3°20′13.36″ | 702 | marl | 22 | organic orchard (cover crop) |
| BL | 37°47′18.37″/3°19′56.31″ | 692 | limestone colluvia | 21 | conventional orchard (bare soil) |
| BM | 37°47′0.89″/3°19′58.41″ | 666 | marl | 21 | conventional orchard (bare soil) |

*2.2. Methods*

2.2.1. Soil Analysis

Soil samples were air dried at room temperature and passed through a 2-mm sieve to obtain the fine-earth (<2 mm) and coarse fragments. All analyses were made on the fine earth fraction. Soil organic carbon (SOC) was determined by digesting the samples, previously ground with a ball mill, with dichromate and sulphuric acid, following the method proposed by Anderson et al. [23]. For other analyses, the procedures specified by the American Society of Agronomy and Soil Science Society of America [24,25] were followed. Total nitrogen (N) was measured with the Kjeldhal method. The pH (1:1) was measured by potentiometry in distilled water. Equivalent $CaCO_3$ was determined by volumetry with a Bernard calcimeter. Exchangeable potassium (K) was extracted with ammonium acetate (pH 7) and determined by flame photometry. Cation exchange capacity (CEC) was determined with the ammonium acetate method (pH 7) and atomic absorption spectrophotometry. Soil contents of clay, silt and sand were estimated with the pipette method, after elimination of organic matter with $H_2O_2$ and dispersion with sodium polyphosphate. The Aggregate Stability Index (ASI) of soils were calculated according to Kemper and Rosenau [26] by means of a wet sieving apparatus (Eijkelkamp Agrisearch Equipment, Zevenaar, The Netherlands).

2.2.2. Mineralogical Analysis

Organic matter was removed with 15% $H_2O_2$ solution on a hot plate and dispersed with a 10% sodium polyphosphate solution. Samples were not treated to remove carbonates because these were measured later. Once the sample was dispersed, the clay-size (<2 μm) fraction was extracted by sedimentation. For preparation of oriented aggregate mounts in XRD studies, the procedure explained in Calero et al. [27] was followed. Iron oxides were removed from clay by the sodium citrate-dithionite bicarbonate (CDB) method and later dialyzed and air-dried. Free iron oxides in CDB extracts were measured in an Agilent 7900 ICP-mass spectrometer (Agilent Technologies, CA, USA). Samples of disoriented crystal powder from the fine-earth fraction (<2 mm) were prepared with a holder filled from the side and scanned from 2 to 50° 2θ in a Siemens D5000 diffractometer (Siemens, Munich, Germany), employing Co Kα radiation (λ = 1.79062 nm). Mineral percentages of disoriented mounts were estimated by intensity-factor methods, as specified by Calero et al. [28].

For the study of phyllosilicates in clay fraction, oriented aggregates of Mg- and K-saturated samples (three saturations with 1 N magnesium chloride and five with potassium chloride, respectively) were prepared by sedimentation and drying on glass slides. The analyses of the slides were carried out after (i) air-drying, (ii) ethylene-glycol (EG) vapour treatment (Mg samples), and (iii) one hour of heating at 550 °C (K samples), according to Calero et al. [27]. In all cases, the Siemens D5000 diffractometer unit was employed under the following operating conditions: Selected Cu K$\alpha_1$ radiation using a germanium-crystal monochromator ($\lambda$ = 0.15406 nm) at 35 kV and 15 mA from an angular range of 2–30° 2$\theta$, with 0.04° 2$\theta$ steps and 2 s counting-time per step. Decomposition of ethylene-glycol (Mg-saturated) patterns in the 3–10° 2$\theta$ zone was performed with the DecompRX software [17], following recommendations made by Barré et al. [29]. Once decomposition and identification of phyllosilicates was complete, the 'relative peak area', called *rpa* (expressed as %), was estimated [30].

### 2.2.3. Specific Surface Area (SSA) Determination of Clays

Nitrogen ($N_2$) surface-area measurements of the clay fraction were done with a Micromeritics surface-area analyzer (Model Gemini 2360, Micromeritics, Norcross, GA, USA), which calculates the specific surface area of clays (SSA) according to the Brunauer–Emmett–Teller (BET) equation.

### 2.2.4. Soil Organic Matter Fractionation

Separation of the various C pools was accomplished by a combination of physical and chemical fractionation techniques in a three-step process, modified from Stewart et al. [31]. After a first step, consisting of the partial dispersion and physical fractionation of the soil to obtain three size fractions (>250 μm, 53–250 μm and <53 μm), a second step, which involved further fractionation of the 53–250 μm fraction previously isolated, was followed. The >250 μm, 53–250 μm and <53 μm fractions isolated after the first step corresponded to the coarse non-protected particulate organic matter (cPOM, hereafter), microaggregate (μagg) and easily dispersed silt and clay (dSilt + dClay) fractions, respectively. In the second step, a density flotation with 1.85 g cm$^{-3}$ sodium chloride was used to isolate fine non-protected POM (LF). After removing the fine non-protected POM, the heavy fraction was dispersed overnight by shaking with 15 glass beads and passed through a 53 μm sieve, separating the microaggregate-protected (>53 μm in size, iPOM) from the microaggregate-derived silt-plus-clay-sized fractions (μSilt+μClay). The third step involved the acid hydrolysis of each of the isolated silt+clay-sized fractions. The silt+clay-size fraction from both the density flotation (μSilt+μclay) and the initial dispersion and physical fractionation (dSilt+dClay) were subjected to acid hydrolysis as described by [9]. Acid hydrolysis consisted of incubating the samples at 95 °C for 16 h in 25 mL of 6 M HCl. After hydrolysis, the suspension was filtered and washed with deionized water over a glass-fiber filter. Residues were oven-dried at 60 °C and weighed. These fractions represent the non-hydrolysable C fractions (NH-dSilt+dClay, hereafter NH; and NH-μSilt+μClay, hereafter μNH). The hydrolysable C fractions (H-dSilt+dClay, hereafter H; and H-μSilt+μClay, hereafter μH) were determined by the difference between the total organic C content of the fractions and the C contents of the non-hydrolysable fractions. This three-step process isolated a total of seven fractions, where SOC content was determined by the Anderson and Ingram procedure [23], assuming the link between the isolated fractions and the protection mechanisms involved in the stabilization of organic C within that pool (described in detail by Six et al. [5]). In particular, the pools and fractions were: the non-protected pool (NPP), composed by the cPOM and LF fractions; the physically protected pool (PPP), composed by μNH, μH and iPOM fractions; the chemically protected pool, equal to the H fraction; and the biochemically protected pool, equal to the NH fraction. Results are expressed in g C in each fraction, regarding the total SOC recovered in the sample (%).

2.2.5. Statistical Methods

A Kolmogorov–Smirnov test was applied to check normality of each variable, transforming by means of simple mathematics when needed. Correlation between normalized variables were analyzed with Pearson's correlation coefficient ($r$), assessing the first-order partial correlations to examine any influence of some relevant soil properties in the relationships of functional and mineralogical fractions. However, our intention was to extract direct correlations between pairs of variables (the partial correlation matrix), taking into account not only the influence of a third variable, but of all others. We considered the whole system as a network [32].

Partial correlation networks (PCN), also known as Gaussian graphical models, allow a user-friendly graphical modelling of the partial correlation matrix in the form of net-like association structures, which reflect causal relationships between variables. Here, nodes represent variables and edges (links connecting two nodes) mean partial correlation coefficients between nodes, where the significance can be assessed by means of statistical testing [33]. The PCN algorithms are only applicable if the sample size $N$ is larger than the number of variables $p$ (more details in Schäfer and Strimmer [33]). As we had more variables than samples, the conventional PCN models could not be applied. Opgen-Rhein and Strimmer [18] introduced a novel approximation to overcome this situation, which is implemented in the "GeneNet" R package. This method applies new estimators to the partial correlation matrix that are stabilized by multiple tests, allowing one to choose a PCN adapted to these kind of data using the false discovery rate as selection criterion [34]. Moreover, GeneNet algorithms can detect the directionality of causal dependences between nodes, if any, generating partially directed networks. In this case, edges would show the direction of causal influence between nodes.

The operational procedure employed by GeneNet can be summarized in three steps [18,33], namely: (1) all possible pairwise correlations between variables (edges) were obtained; (2) the edges were listed according to the correlation coefficient value, from highest to lowest; and (3) the $p$-values and the false discovery rate (*fdr*) were specified for each one. From the edge output list, only those that exceeded a threshold of significance and *fdr* were taken to build the graph. The directionality of each edge was tried by the *fdr* of the corresponding test at the same threshold values. Finally, significant edges were exported to SocNet software and plotted as a weighted network, including directionality as arrowed edges if any causal dependence was detected in the test.

## 3. Results

### 3.1. Soil Properties

The values of the physical and chemical properties indicate that studied soils that are poorly suited for agriculture, with high proportion of coarse fragment (>20%), especially in soils over the acidic granodiorite parent materials (Table 2).

As expected, properties of soils under the contrasted parent materials widely differed. Soil pH, equivalent $CaCO_3$, CEC and exchangeable potassium were higher in soils under calcareous parent materials. Management had a great influence in those properties related to soil organic matter (SOM). SOC, total N and available K were typically higher in the woodland soils and lower in the bare soils of the olive groves, with intermediate values in soils with cover crops (Table 2). The highest structural stability (0.92 on average) was found in the woodland soils whereas the values in the bare soils and soils with cover crops were quite similar (0.78 and 0.80 on average, respectively). Calcareous soils showed greater specific surface area but lower $Fe_2O_3$ contents than the siliceous (Table 2). $Fe_2O_3$ contents tended to be lower under cover crops in both calcareous and siliceous soils.

**Table 2.** Selected physical and chemical soil properties of the sites.

| Sample | CF (%) | Sand (%) | Clay (%) | ASI * (%) | SOC (%) | Total N (%) | pH | CaCO₃ (%) | K (ppm) | CEC (cmol + kg⁻¹) | SSA_clay (m² g⁻¹ clay) | Fe₂O₃clay (mg g⁻¹ clay) |
|---|---|---|---|---|---|---|---|---|---|---|---|---|
| WG | 80 | 59 | 10 | 0.94 | 10.51 | 0.4 | 6.18 | 2 | 210 | 17.40 | 1.4 | 103 |
| BG | 77 | 49 | 27 | 0.79 | 0.81 | 0.04 | 5.88 | 2 | 39 | 12.80 | 1.3 | 92 |
| CG | 54 | 57 | 12 | 0.73 | 1.56 | 0.10 | 6.24 | 1 | 73 | 10.00 | 7.5 | 57 |
| WL | 38 | 38 | 32 | 0.9 | 5.78 | 0.35 | 7.69 | 4 | 631 | 43.60 | 28.7 | 47 |
| CL | 28 | 37 | 28 | 0.92 | 4.23 | 0.33 | 7.79 | 22 | 638 | 35.20 | 24.6 | 63 |
| CM | 45 | 34 | 25 | 0.75 | 1.87 | 0.11 | 8.00 | 46 | 136 | 22.40 | 8.6 | 33 |
| BL | 26 | 37 | 30 | 0.96 | 1.94 | 0.11 | 7.96 | 51 | 232 | 25.40 | 30.0 | 36 |
| BM | 49 | 25 | 41 | 0.59 | 1.11 | 0.06 | 8.04 | 57 | 118 | 18.80 | 9.9 | 6 |

W stands for forest; B and C for olive groves with bare soil or with spontaneous cover crops, respectively. G, L and M stand for siliceous and calcareous limestones or calcareous marl, respectively. CF: coarse fragments; SOC: soil organic carbon; CEC: cation exchange capacity; Total N: total nitrogen, CaCO₃ eq: equivalent calcium carbonate; K: exchangeable potassium, SSA_clay: specific surface area of clays; Fe₂O₃clay: citrate-dithionite-bicarbonate free iron of clay. * ASI: aggregate stability index = A/B × 100, A: weight of soil stable macroaggregates (≥0.25 mm) resisting wet sieving, B: weight of soil unstable macroaggregates (<0.25 mm) passing through the 0.25 mm grid.

Soil mineralogy of fine earth matched the bedrock composition (Table 3). Broadly, granodiorite soil samples were richer in quartz than calcareous samples, while phyllosilicates tended to predominate in the latter. Calcite was close to the analytical amount of carbonates. In the calcareous sites, the carbonates varied according to both the parent material and the soil management, being larger in the marls than in limestone colluvia, and larger in the bare soils than in the soils with cover crops. In the siliceous site, mineral composition was more homogeneous, although soils with cover crops showed less quartz (37%) and more phyllosilicates (53%) than the other granodiorite soils. Other minerals (oxides, feldspar and chlorite) were scarce and did not show a clear trend among soils.

**Table 3.** Mineralogical composition of the fine-earth fraction (<2 mm; DRX- disoriented powder. %) of the sites.

| Sample | Phyll. | Q. | Goe. | Hem. | Chl. | FdK | FdCa-Na | Calcite | Dol. |
|---|---|---|---|---|---|---|---|---|---|
| WG | 31 | 61 | 3 | tr. | 2 | 1 | 1 | tr. | tr. |
| BG | 32 | 62 | 3 | tr. | 2 | 1 | 1 | tr. | tr. |
| CG | 53 | 37 | 3 | 1 | 1 | 1 | 3 | tr. | tr. |
| WL | 61 | 24 | 3 | 1 | 2 | 2 | 3 | 2 | 1 |
| CL | 54 | 17 | 2 | tr. | 2 | 1 | 1 | 23 | tr. |
| CM | 39 | 13 | 2 | tr. | 5 | 1 | 1 | 38 | 1 |
| BL | 38 | 12 | 2 | tr. | 2 | 1 | 1 | 43 | 1 |
| BM | 31 | 7 | 1 | tr. | 2 | 1 | tr. | 57 | 1 |

W stands for forest; B and C for olive groves with bare soil or with spontaneous cover crops, respectively. G, L and M stand for siliceous and calcareous limestones or calcareous marl, respectively. Phyll: Phyllosilicates; Q: quartz; FdK: potassium feldspar; FdCa-Na; calcium–sodium feldspar; Goet: goethite; Hem: haematite; Cal: calcite; Dol: dolomite. tr: traces (<1%).

### 3.2. Clay Fraction

Clay fractions (Figures 1 and 2, Table 4) were composed mainly of 2:1 phyllosilicates, which often showed poor swelling behavior in Mg-glycol treatment. The most abundant minerals were illite, vermiculite and various types of mixed layers.

Illite was detected in every sample as a peak near 1.000 nm (0.990 to 1.019 nm) with second- and third-order reflections at 0.500 and 0.330 nm. Its contents varied widely from less than 10% (BM) to more than 70% (WL). The 1.000 nm-spacing mica peaks were fitted by decomposition (Figures 3 and 4) to two curves of differing crystallinity and angular positions, corresponding to different phases: poorly crystallized illites (PCI, peak width at half height, WHH, >0.4° 2θ) and well crystallized illites (WCI) [35].

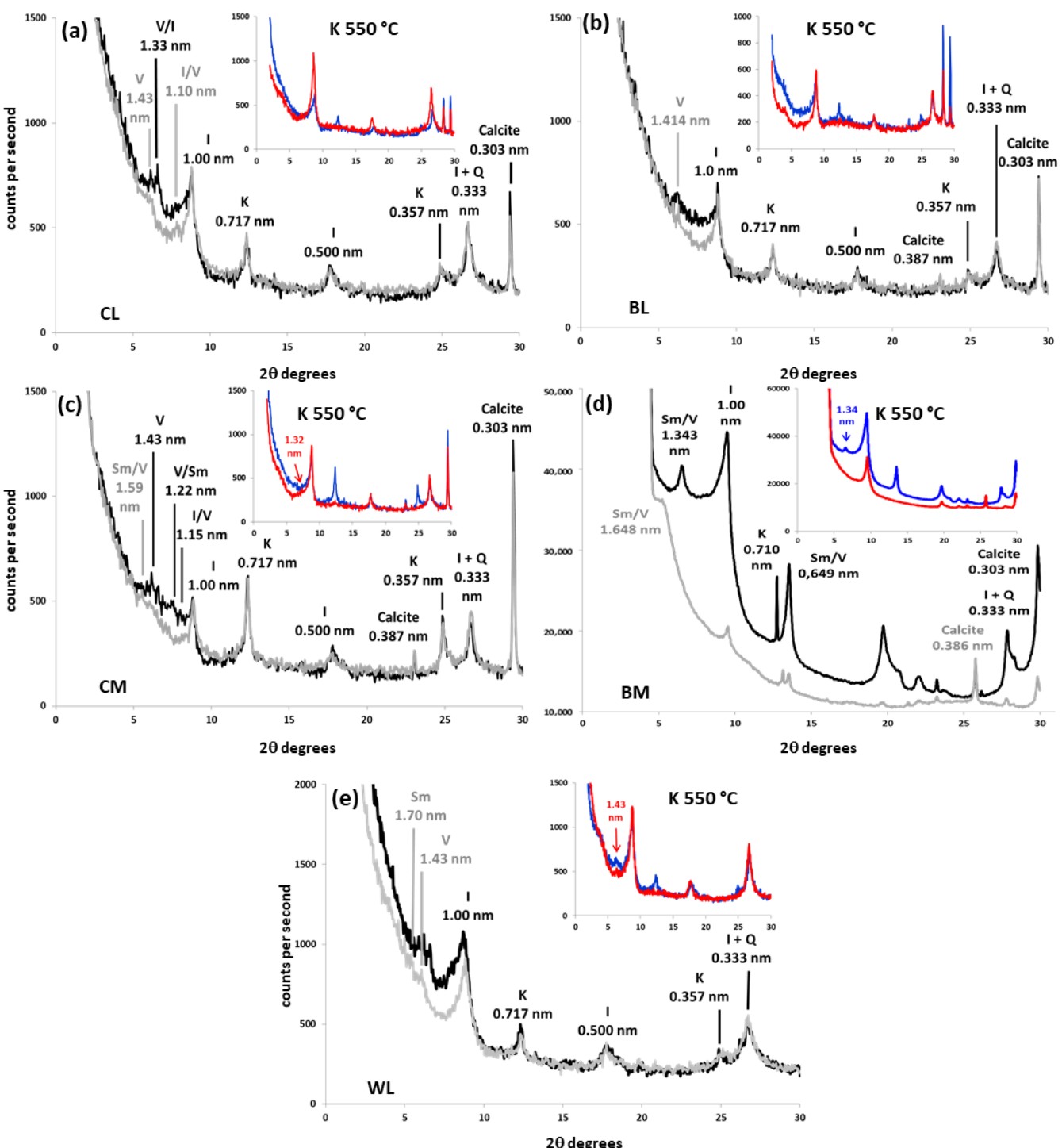

**Figure 1.** XRD patterns of oriented mounts (3–30 °C 2θ) from clay fraction of the calcareous soils with different treatments (black: Mg-saturated and air dried; grey: Mg-saturated and EG-solvated; blue: K-saturated and air dried; red: K-saturated and heated at 550 °C). (**a**) Cover-crop soil over limestone colluvia (CL); (**b**) bare soil over limestone colluvia (BL); (**c**) cover-crop soil over marls (CM); (**d**) bare soil over marl (BM); (**e**) woodland soil over limestone colluvia (WL). Q: quartz; Sm: smectite; Sm/V: smectite–vermiculite mixed layer; V/Sm: vermiculite–smectite mixed layer; V: vermiculite; V/I: vermiculite–illite mixed-layer; I/V: illite–vermiculite mixed-layer; I: illite; K: kaolinite.

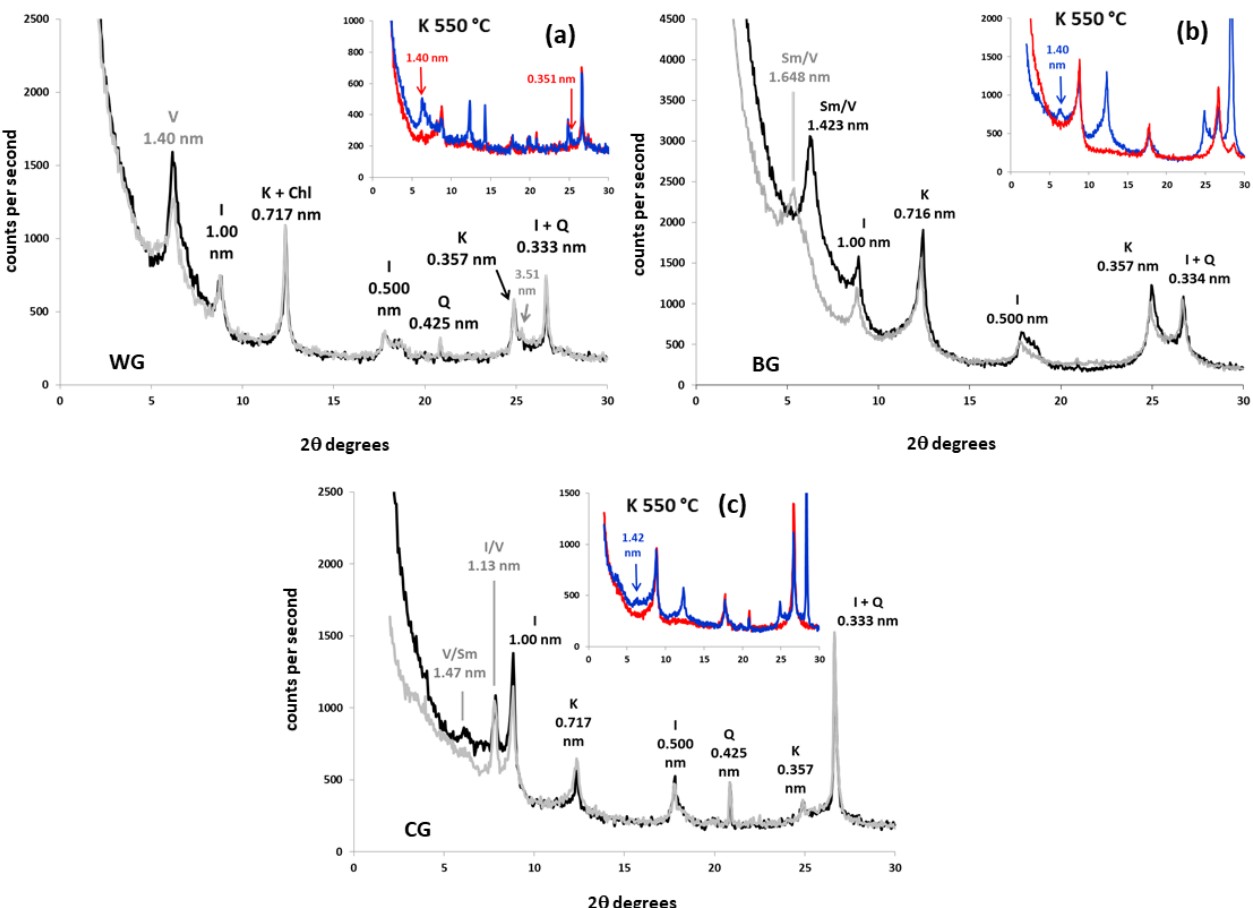

**Figure 2.** XRD patterns of oriented mounts (3–30 °C 2θ) from clay fraction of the siliceous soils with different treatments (black: Mg-saturated and air dried; grey: Mg-saturated and EG-solvated; blue: K-saturated and air dried; red: K-saturated and heated at 550 °C). (**a**) Woodland soil over granodiorite (WG); (**b**) bare soil over granodiorite (BG); (**c**) cover-crop soil over granodiorite (CG). Q: quartz; Sm/V: smectite–vermiculite mixed-layer; V/Sm: vermiculite–smectite mixed layer; V: vermiculite; I/V: illite–vermiculite mixed-layer; I: illite; Chl: chlorite; K: kaolinite.

**Table 4.** Mineralogical composition of clay fraction (DRX- oriented aggregate. Mg-EG. % relative peak area, rpa).

| Sample | Sm. | V. | Illite | ML | MLV [1] | MLS [1] | Chl. | K | FdK | FdCa-Na | Q. | Calcite |
|--------|-----|-----|--------|-----|-----|-----|------|-----|-----|---------|-----|---------|
| WG | 1 | 45 | 15 | 3 | 3 | 0 | 10 | 22 | 1 | 1 | 2 | 0 |
| BG | 0 | 12 | 10 | 43 | 1 | 42 | 5 | 28 | 1 | 0 | 1 | 0 |
| CG | 2 | 2 | 37 | 28 | 28 | 0 | 5 | 20 | 1 | 1 | 4 | 0 |
| WL | 1 | 6 | 72 | 2 | 0 | 2 | 5 | 7 | 4 | 2 | 1 | 0 |
| CL | 1 | 5 | 49 | 5 | 4 | 1 | 6 | 13 | 2 | 1 | 1 | 17 |
| CM | 1 | 3 | 20 | 5 | 1 | 4 | 5 | 23 | 1 | 1 | 2 | 39 |
| BL | 0 | 2 | 42 | 4 | 4 | 0 | 7 | 13 | 2 | 1 | 1 | 28 |
| BM | 7 | 0 | 7 | 13 | 0 | 13 | 4 | 3 | 6 | 3 | 1 | 56 |

W stands for forest; B and C for olive groves with bare soil or with spontaneous cover crops, respectively. G, L and M stand for siliceous and calcareous limestones or calcareous marl, respectively. Q: quartz; FdK: potassium feldspar; FdCa-Na; calcium-sodium feldspar; Sm: smectite; V: vermiculite; ML: mixed layers; MLV: vermiculite mixed-layer; MLS: smectite mixed-layer; Chl.: chlorite. K: kaolinite. [1] The relative distribution between smectite and vermiculite mixed-layer, regarding the total ML measured in the 3–30 °C 2θ oriented mounts, was carried out from the relative peak areas of Sm/V (smectite mixed layers, MLS) vs. V/Sm + I/V + V/I (vermiculite mixed layers, MLV) in the respective decomposed patterns.

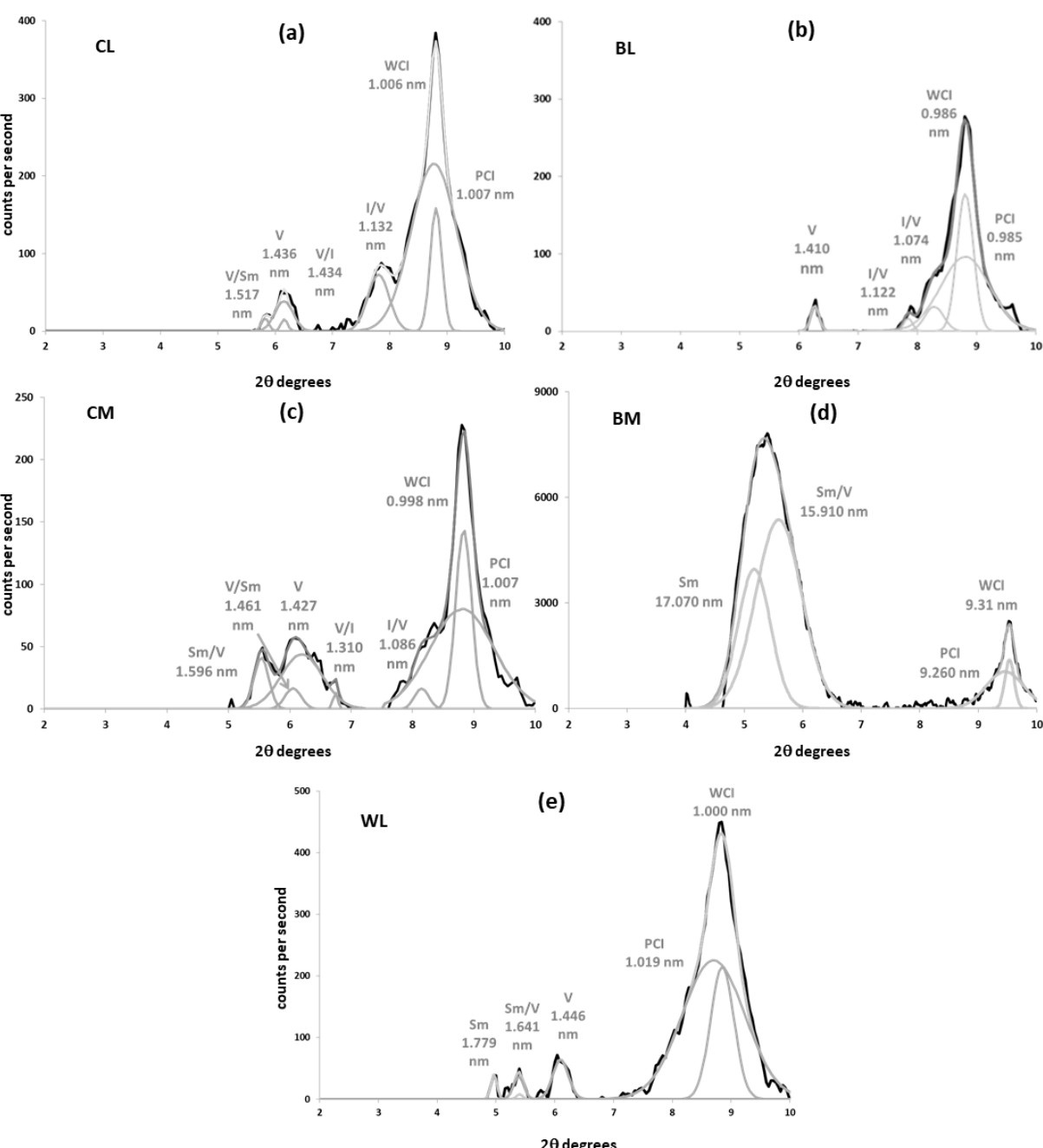

**Figure 3.** Peak decomposition of XRD patterns of oriented mounts (3–10° 2θ) from clay fraction of the calcareous soils (Mg-saturated and EG-solvated). Black: experimental pattern; Grey: simulated pattern. (**a**) Cover-crop soil over limestone colluvia (CL); (**b**) bare soil over limestone colluvia (BL); (**c**) cover-crop soil over marls (CM); (**d**) bare soil over marl (BM); (**e**) woodland soil over limestone colluvia (WL). Sm: smectite; Sm/V: smectite–vermiculite mixed-layer; V/Sm: vermiculite–smectite mixed layer; vermiculite; V/I: vermiculite–illite mixed-layer; I/V: illite–vermiculite mixed-layer; V: vermiculite; PCI: poorly crystalized illite; WCI: well-crystalized illite.

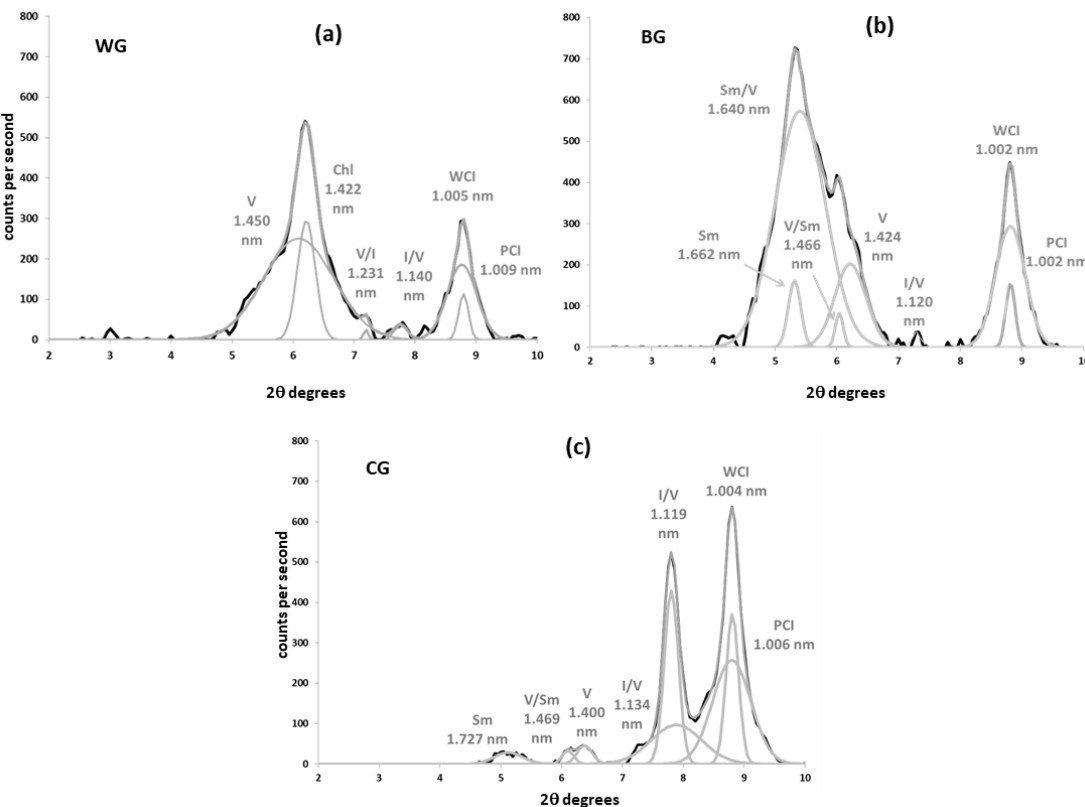

**Figure 4.** Peak decomposition of XRD patterns of oriented mounts (3–10° 2θ) from clay fraction of the siliceous soils (Mg-saturated and EG-solvated). Black: experimental pattern; Grey: simulated pattern. (**a**) Woodland soil over granodiorite (WG); (**b**) conventional soil over granodiorite (BG); (**c**) cover crop soil over granodiorite (CG). Sm: smectite; Sm/V: smectite–vermiculite mixed-layer; V/Sm: vermiculite–smectite mixed layer: vermiculite; V/I: vermiculite–illite mixed-layer; I/V: illite–vermiculite mixed-layer; V: vermiculite; Chl.: chlorite; PCI: poorly crystalized illite; WCI: well-crystallized illite.

Soil vermiculites are characterized by a ≃1.420 nm peak that tends to shift to 1.020 nm under $K^+$-saturation [36,37]. This was the most common behavior in calcareous soils (Figure 1). However, the interlayer collapse seemed only partial in granodiorite samples and in the woodland colluvium sample (WL), which retained ≃1.420 nm peaks of variable intensity in potassium solutions (Figures 1 and 2). According to Douglas [36], the inhibition of potassium fixation and the partial shift are caused by the presence of non-exchangeable hydroxyl-Al in vermiculite interlayers. The greater the hydroxylation of the interlayer, the smaller the shift of the 1.420 nm peak. When the interlayer is completely filled with non-exchangeable hydroxides, the 001 peak remains at about 1.420 nm under $K^+$-saturation; we identified these as soil chlorites [30,38]. This mineral seems to be abundant in the woodland granodiorite soil (WG, Figure 2a, Table 4): The 001 peak did not expand in glycol or collapse noticeably in potassium solution, maintaining the same height and area relation with the mica peak (1.000 nm) as in magnesium treatments. Furthermore, it kept the ≃1420 nm reflection in the heat treatment, although much of it partially collapsed, generating a wide band between 1.420 and 1.000 nm. Decomposition of the 1.400 nm peak from WG (Figure 4a) showed two curves with different crystallinities, the main of which were identified as soil vermiculite (1.450 nm) and the other, more crystalline, as soil chlorite (1.422 nm).

In some XRD patterns from acid (BG, Figure 2b) and calcareous soils (CM, BM, WL, Figure 1c–e, respectively), the air-dried magnesium ≃1.420 nm peaks expanded to positions closer to 1.700 nm upon glycol treatment. These peaks appeared between ≃1.600 and 1.650 nm in decomposition patterns (Figures 3c–e and 4b). This should be attributed to

smectite–vermiculite mixed layers [39], which were detected very often, together with the hydroxyl-interlayered phases, in soils [30]. As soil vermiculites, they tend to collapse at ≃1.000 nm with saturation in potassium at 550 °C (Figures 1 and 2). When this mixed-layer did not swell over 1.600 nm and was kept close to the vermiculite (between ≃1.420 and 1.600 nm), it was attributed to vermiculite–smectite mixed-layer. It is the same phase as smectite–vermiculite, but with a relatively higher content of vermiculite.

Finally, several bands were detected in the sector 7.7–6.3° 2θ (1.000–1.400 nm spacing) of magnesium patterns. The soil with cover crops over granodiorite (CG) was the sample where this phase seemed more conspicuous (Figures 2c and 4c). These reflections, which did not swell appreciably in glycol and collapsed at 1.000 nm in K 550 °C, were attributed to interstratified vermiculite–illite according to Sawnhey [40]. Decomposition confirmed the presence of these phases, which appeared with relatively low crystallinity (very wide compared to their height) and in low quantities, except in CG, where it is the most prevalent phyllosilicate after illite.

In addition to the 2:1 phyllosilicates, other phases were detected in the clay fraction. As expected, calcareous soils displayed a calcite peak at 0.303 nm, except for soil from the woodland over limestone colluvia where there was a lack of calcite. Calcite in the <2 μm fraction was also larger in marls than in their equivalent soil over limestone colluvia (CM > CL, BM > BL). Kaolinite was also identified (001 reflections at ≃0.714 nm and 002 at ≃0.358 nm). It was a fairly prevalent mineral, reaching more than 20% in the samples over granodiorites and also in the organic orchard over marls. Other silicate minerals, such as quartz (0.426 nm) or feldspars (0.325–0.320 nm), which were abundant in the fine earth, were below 5% in the clay fraction.

### 3.3. Soil Organic Matter: Functional Pools

The non-protected SOC pool, mostly formed by the coarse particulate organic matter (cPOM), was the main pool (29.2–69.9%) in most soils (Table 5). The average value in the woodland sites was typically >60%, whereas it only reached 35%, on average, in the bare soils and 45% in the olive groves with cover crops.

**Table 5.** Contribution (%) of fractions and sub-fractions to the total soil organic carbon.

| Sample | Non Protected Pool | | | Chemically Protected | Biochemically Protected | Physically Protected Pool | | | |
|---|---|---|---|---|---|---|---|---|---|
| | cPOM | LF | Total | H | NH | μNH | μH | iPOM | Total |
| WG | 53.56 | 16.39 | 69.95 | 5.34 | 7.54 | 5.56 | 2.60 | 9.02 | 17.18 |
| BG | 24.79 | 4.79 | 29.58 | 17.54 | 18.70 | 20.82 | 9.26 | 4.11 | 34.19 |
| CG | 25.41 | 3.79 | 29.2 | 14.36 | 16.52 | 16.50 | 7.99 | 15.43 | 39.92 |
| WL | 43.29 | 8.37 | 51.66 | 11.58 | 10.61 | 7.24 | 7.35 | 11.56 | 26.15 |
| CL | 59.46 | 2.85 | 62.31 | 8.82 | 5.33 | 5.27 | 8.00 | 10.26 | 23.53 |
| CM | 43.42 | 0.00 | 43.42 | 15.48 | 7.06 | 6.05 | 15.23 | 12.77 | 34.05 |
| BL | 31.7 | 5.03 | 36.73 | 19.84 | 9.49 | 6.05 | 15.53 | 9.13 | 30.71 |
| BM | 34.89 | 3.92 | 38.81 | 19.73 | 8.12 | 6.05 | 17.60 | 10.07 | 33.72 |

W stands for forest, whereas B and C for olive groves with bare soil or with spontaneous cover crops, respectively. G, L and M stands for siliceous and calcareous limestones or calcareous marl, respectively. NH: Non-hydrolysable organic carbon, biochemically protected; μNH: Non-hydrolysable organic carbon, physically protected; H: Hydrolysable organic carbon, chemically protected; μNH: Hydrolysable organic carbon, physically protected; LF = light fraction, non-protected; iPOM: occluded particulate organic matter, physically protected; cPOM: coarse particulate organic matter, non-protected; P-P: protected pool.

In general, the physically protected SOC was the second most-abundant SOC pool. This contributed to an average of 33% in the bare soils, a figure similar to that of the non-protected pool. Moreover, this pool was the most abundant in the granodiorite cultivated soils (BG: 34%; CG: 40%). Within this pool, the occluded particulate carbon (iPOM) was lower than the chemical (μH) and biochemical-microaggregate (μNH) subfractions, particularly in bare soils, where it varied similarly to the coarse particulate carbon (cPOM).



Samples from non-carbonated soils, including the woodland soil over colluvium (WL), showed quite similar average values of the chemically and biochemically protected pools, with 12–13% of SOC. However, in carbonated soils, the percentage of the chemically protected pool (15%) almost doubled that of the biochemically protected pool (8%). The <53 μm hydrolysable fraction from microaggregates (μH) behaved like the chemically protected pool, having larger mean percentages in carbonated (13%) than in non-carbonated samples (7%). Likewise, it seems to be an effect of soil management in the total hydrolysable pool, including the chemically protected-plus-μH fractions. In relative terms, this pool was more abundant in bare soils (average of 33% of total SOC) than in groves with cover crops (23%) and woodland (13%). These results may be explained because bare soils are, on average, richer in carbonates (Table 2), which promotes the organo-humic complexes formation [12]. Lastly, the biochemically protected pool followed the opposite trend to the chemically protected pool, being higher in siliceous soils. With an average value <10%, this fraction was the least abundant of the SOC pools.

*3.4. Relationships between Functional Fractions and Clay Mineral Assemblages*

Pearson's correlation matrix among clays and functional pools is showed in Suplementary materials Tables S1–S4. Some key physical and chemical properties (e.g., SOC, texture and CEC) were also introduced in the correlation analysis because they appeared markedly conditioned by the parent material and soil management.

Overall, a scarce number of significant correlations were found, especially between the two types of properties. This was not entirely unexpected, given the relatively low harmonization between the employed methodologies. The variable with the highest number of significant correlations was the SOC. The SOC was directly related to the non-protected SOC pool (NPP: cPOM and LF) and inversely to the chemically and physically protected SOC (H and PPP, respectively). In addition, SOC content was significantly positively correlated with soil chlorite, but negatively correlated with mixed layers. Given its weight in the correlation matrix, SOC may have a decisive effect in reducing the partial *r* and statistical significance for other correlations between mineral and functional fractions. Accordingly, this effect was explored through a partial correlation analysis (Table 6).

**Table 6.** Pearson's correlation coefficients (*r*) for soil organic carbon fractions and the mineral phase, controlling the effects of soil organic carbon.

| SOC Fraction | >2 mm Fraction | *r* | Partial *r* | *p*-Value |
|:---:|:---:|:---:|:---:|:---:|
| NH | ML | 0.810 * | 0.761 | 0.047 |
| μNH | ML | −0.794 * | −0.711 | 0.073 |
| TNH | ML | 0.829 * | 0.782 | 0.038 |
| cPOM | ML | −0.727 * | −0.415 | 0.355 |
| NPP | ML | −0.728 * | −0.415 | 0.521 |
| PPP | Chl | −0.743 * | −0.240 | 0.605 |

Significance: * < 0.05; TNH: total non-hydrolysable organic carbon (NH plus μNH); NH: Non-hydrolysable organic carbon. biochemically protected. μNH Non-hydrolysable organic carbon. physically protected. ML: mixed layers. Chl: chlorite; cPOM; coarse particulate SOC; NPP: non-protected-SOC pool; PPP: physically protected-SOC pool.

Interestingly, the second variable with the highest number of correlations was the content of mixed layers (ML). It was negatively correlated with the non-protected pool and the cPOM fraction (*r* = −0.728 and −0.727, respectively; *p* < 0.05). Both correlations might be explained by their high correlation with SOC; therefore, the partial correlation between mixed-layer, cPOM and NPP by controlling total SOC became statistically not significant (partial *r* = −0.415, *p* = 0.355, and partial *r* = −0.415, *p* =0.521, respectively, Table 6).

Despite this, the correlation of the mixed layers with the biochemically protected pool and the total non-hydrolysable fraction TNH (sum of the former plus μNH), did not lose statistical significance when controlling for the effect of SOC (partial *r* = 0.761 for NH and 0.782 for TNH, *p* <0.05, respectively), and almost kept the significance at 0.05 with the

fraction μNH (partial *r* = −0.711, *p* = 0.073; the negative sign of *r* is due to the inverse transformation required for normalization (Table 6, Figure 5).

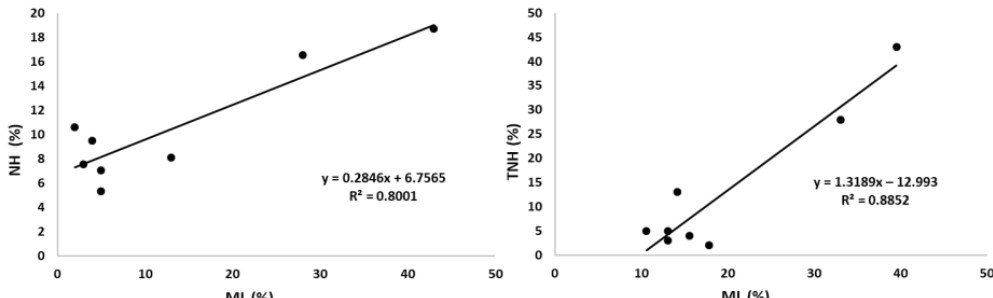

**Figure 5.** Correlations between the non-hydrolysable fractions and the mixed layers. NH: non-hydrolysable fraction, biochemically protected; TNH: NH plus μNH (non-hydrolysable fraction, physically protected); ML: mixed layers.

The calcite in clay fraction also showed a relatively high number of correlations, although the only organic fraction that correlated significantly was the hydrolysable fraction from microaggregates μH (*r* = 0.885, *p* = 0.004, Table 7, Figure 6) and the total hydrolysable pool TH, defined as the sum of the chemically protected plus the μH fractions (*r* = 0.748, *p* = 0.033). However, no relationship between the chemically protected fraction and calcite in clay was found. Calcite was strongly and positively correlated with soil pH (*r* = 0.768, *p* = 0.026), but negatively with sand (*r* = −0.828, *p* = 0.011) and % Fe$_2$O$_3$ in clay (*r* = −0.818, *p* = 0.013). As soil pH correlates simultaneously with μH and the calcite in clay, we tested the partial correlation between both fractions controlling the soil pH (Table 7), giving that statistical significance is barely maintained for the correlation between calcite and μH.

**Table 7.** Pearson's correlation coefficients (*r*) for soil organic carbon fractions and the mineral phase, controlling the effects of pH.

| SOC Fraction | >2 μm Fraction | *r* | Partial *r* | *p*-Value |
|---|---|---|---|---|
| μH | Calcite | 0.885 ** | 0.753 | 0.050 |
| TH | Calcite | 0.748 * | 0.662 | 0.136 |

Significance: * < 0.05. ** < 0.01. TH: total hydrolysable organic carbon (μH plus H); μH: hydrolysable organic carbon, physically protected.

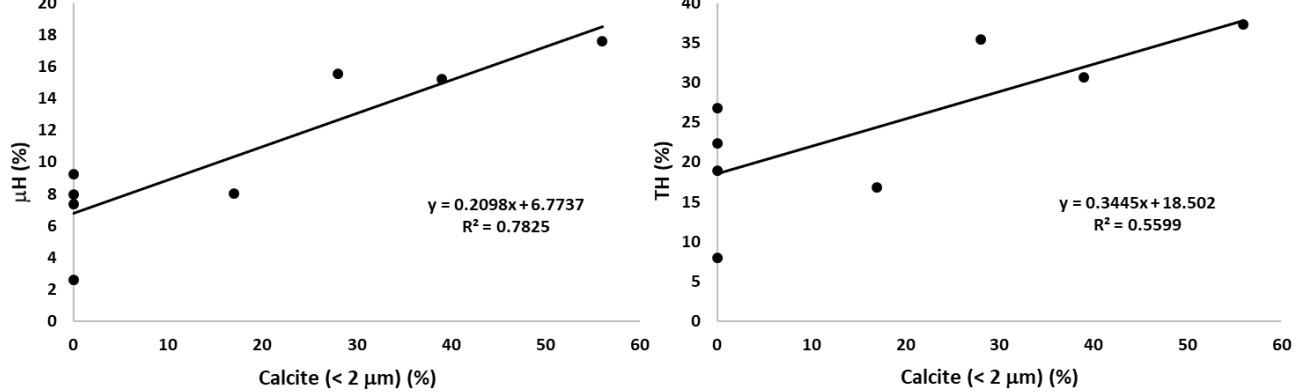

**Figure 6.** Correlations between the hydrolysable fractions and the calcite in clay. μH: hydrolysable fraction, physically protected; TH: H plus μH.

### 3.5. Partial Correlation Networks

To represent the direct relationship structure amongst carbon pools (NPP, PPP, biochemically and chemically protected) and fractions (cPOM, LF, NH, H, iPOM, μNH, μH), and the clay mineral phases, different weighted partial-correlation networks (PCN) were

obtained. Given the high number of correlations of total soil carbon in the correlation matrix, the SOC was included in the analysis to avoid losing the statistical significance of the PCNs [32].

The PCN from the four total carbon pools (NPP, PPP, biochemically and chemically protected) were scarcely significant, since all their nodes had a false discovery ratio (*fdr*) >0.40. However, we obtained a significant PCN from the seven SOC subfractions (Figure 7) at the useful threshold of *fdr* < 0.2 and *p* < 0.05 [41].

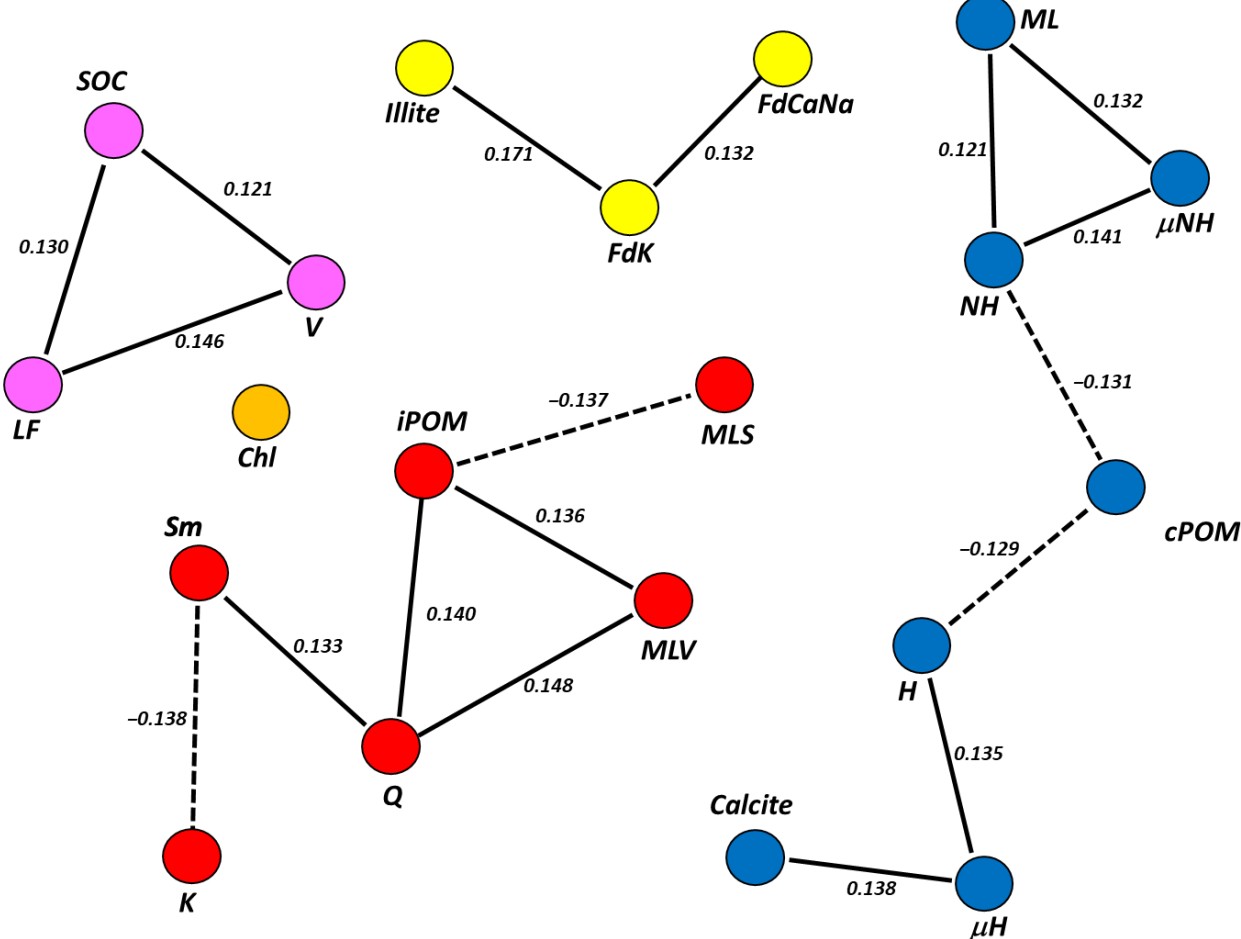

**Figure 7.** PCN between SOC pool subfractions and soil clay minerals at *p* < 0.05 and false discovery rate < 0.20. Dashed lines: negative correlations. (For abbreviations, see Tables 2–5). Colors show different clusters of variables.

Although significant, this PCNs showed weak direct correlation coefficients between nodes (*r* < 0.180). Since partial correlations can be interpreted as coefficients in a linear regression model, its weakness indicates a low (but significant) predictive power for the linked variables [32]. Directionality tests for edges indicated both PCNs were undirected networks (*p* > 0.05, *fdr* = 1), and thus, the directionality of this predictive relationship—which node would act as independent or dependent variable in a regression—remains unclear [18]. Another remarkable feature of the PCN is that it was formed by clusters of variables, which means sets of nodes connected to each other and isolated from the rest.

In Figure 7, a PCN composed by four clusters could be observed: two clusters of three nodes, one cluster of six and another cluster of seven. Chlorite did not appear connected to any cluster significantly (*fdr* > 0.30). One of the two clusters with the lowest number of nodes was formed solely by SOC, LF fraction and vermiculite, and the other cluster by primary silicates: illite and feldspars. A third cluster was composed of cPOM fractions, which correlated negatively with two groups of variables: (1) the non-hydrolysable fractions

(μNH and NH) and mixed layers (ML), mutually correlated with each other in a positive way; and (2) the hydrolysable fractions (μH and H) and calcite, also positively correlated with each other. In the latter, the correlation between H and calcite was mediated by a third variable, μH. This implies a predictive effect between both variables (H over calcite, or calcite over H, because directionality could not be stablished), but only through μH. One last cluster grouped mainly clay minerals, including the two sub-phases of mixed-layer, MLV and MLS, and quartz. The only organic fraction in this cluster was the iPOM. This fraction had a positive correlation with Q, MLV and smectite (mediated by Q), and a negative correlation with MLS and kaolinite (in this case, mediated by Q and Sm). Here, it could be highlighted that MLV and MLS did not show partial correlation with ML at *fdr* < 0.20. Nevertheless, if the significance of the model is decreased to an *fdr* < 0.25, new edges connecting all clusters in a single, large cluster, emerged (Figure 8), among them, those that connect MLS with ML (partial *r* = 0.110). Other edges that connected the four clusters were the correlation between H and SOC (partial *r* = −0.109) and the correlation between smectite and calcite (partial *r* = 0.106).

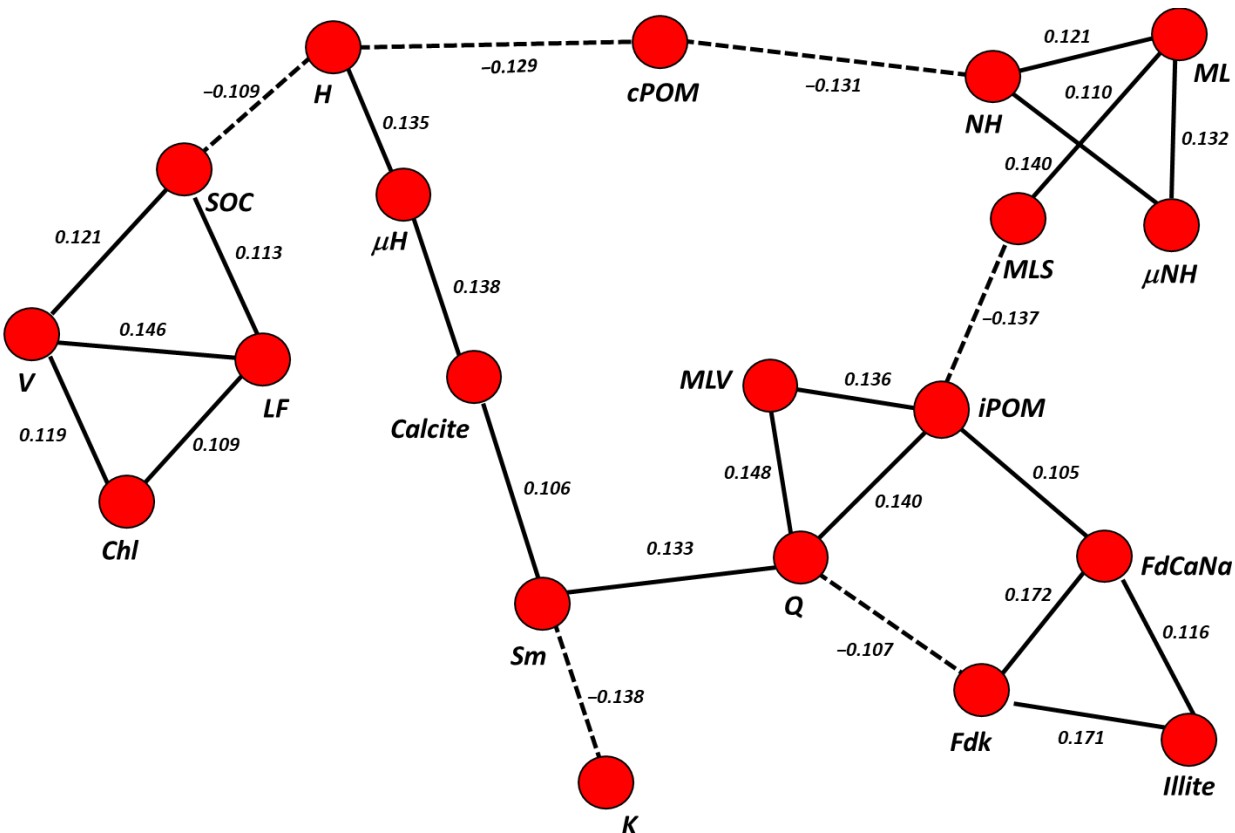

**Figure 8.** PCN between SOC pool subfractions and soil clay minerals at *p* < 0.05 and false discovery rate < 0.25. Dashed lines: negative correlations. (For abbreviations, see Tables 2–5).

## 4. Discussion

The lack of spontaneous cover crops in bare orchards declines SOC with respect to the comparable and nearby forest. Current levels of SOC concentration in the topsoil of the bare orchards was only ca. 23% of the concentrations found in the adjacent and comparable woodland. This reduction is similar, albeit in a lower range, in the comparison of SOC in topsoil among olive orchards with different management and natural areas reported for the region [42]. The increased soil disturbance, the near-total lack of biomass returned to the soil and the higher erosion rate in the bared olive orchard explain this difference. The implementation of spontaneous cover crops improved the level of SOC, not only because the increase of biomass C which enters the soil, but also because of the reduction in soil ero-

sion [21]. Thus, spontaneous cover crops could be a technically and economically strategic way to increase the levels of SOM and SOC in olive orchards. This agrees well with other studies that have reported rates of increase in SOC in olive orchards using conservation-agriculture techniques, such as cover crops and incorporation of organic residues. For instance, in a meta-analysis made by Vicente-Vicente et al. [43], a response ratio was found (the ratio of SOC under spontaneous plant-community management compared with a bare soil managed orchard) from 1.1 to 1.9, suggesting that under conservation management, which included cover crops, the SOC was doubled at a maximum.

Not only total SOC declines in olive groves compared with natural soils, but also the relative contribution of different SOC fractions, which may have important implications for organic matter management. Non-protected fractions showed the largest content in woodland sites, intermediate content in the groves with cover crops, and was the lowest in groves with bare soil. This pattern is consistent with the expected incorporation rate of fresh organic matter in each system. Furthermore, Six et al. [5] stated that larger non-protected SOC levels in plant-covered soils, compared with bare soils, could also be related to the stability of the soil macroaggregates. As a matter of fact, the large differences of ASI between BL and BM (Table 2) may reveal a remarkable effect of the parent material in the structural stability in this parameter. In Plaza et al. [15], it was also stated that a weak structure at the macroaggregate level in marl-based conventional olive grove soils was associated with the physicochemical processes in smectite surfaces.

Differences in the different SOC fractions between the woodland and olive orchards were similar to those described previously when comparing cropland and woodland areas, with the latter presenting a higher concentration of SOC (most of it in the unprotected fraction), while cropland soils presented a higher fraction of carbon in the physically and chemically protected fractions [44]. Most likely, this result could be explained by the fact that, because under soil degradation and lower annual organic carbon inputs, particularly under bare olive orchard, most of the unprotected SOC decomposes relatively quickly and a greater proportion of the remaining SOC is protected. Therefore, the low unprotected SOC concentration found in the bare olive orchard is an issue in the potential increase of SOC stock, as pools of physically, chemically and biochemically protected SOC depend heavily upon the pool of unprotected SOC. Nevertheless, the protected SOC did not show a homogeneous development through olive groves, and different compositions of the same could be found depending on the soil management. Specifically, the physically protected pool was quite similar in both olive groves (and higher to that in the woodland soils). Considering the three subfractions of this pool, the occluded particulate matter (iPOM) was particularly abundant in cover-crop soils, while μNH and μH showed a similar trend to the chemically and biochemically protected SOC, being higher in bare soil. As stated above, the contribution of cPOM was higher for soils under spontaneous plant community with respect to groves with bare soil, fueling the physically protected SOC throughout occlusion or particulate carbon into microaggregates (iPOM). This process highlights that μNH and μH might play a similar functional role in bare and plant-covered soils, beyond that they are included as a part of the physically protected pool.

Overall, the relative SOC distribution in the sites of this study follow good, established conceptual models for different ecosystems [8]: the active or labile/active pool, corresponding to the free particulate SOC (cPOM, LF), is more abundant in forests; the intermediate pool (iPOM), more abundant in non-tilled soils; and the stable/passive pool, corresponding to the surface mineral-linked SOC (H, μH), and the biochemically recalcitrant pool (NH, μNH), enriched in bare and deeply disrupted soils. The distribution among SOC fractions was also similar to the results obtained by Vicente-Vicente et al. [45], who measured SOC fractions distribution in olive oil orchards with temporary cover crops.

The composition of the clay fraction reflected, to a great extent, the parent material of soils. Granodiorites are richer in soil vermiculites, either purely or forming part of mixed layers (vermiculite–illite, vermiculite–smectite). In Parizec and Girty [46], it was reported that the occurrence of soil vermiculites was a weathering product of biotite and muscovite

from crystalline residua of granodiorites in temperate climates. Moreover, they can be associated to hidroxyl-Al minerals in moderately acidic soils, due to the precipitation of hydroxides in the interlayer [38]. On the other hand, the insoluble residue from limestones is mainly composed by illite, while smectite is the phyllosilicate dominant in marls in southeastern Spain [47]. However, some trends related to soil management could be also stated. In general, soil samples from bare olive groves showed more swelling and smectite layers than those samples from cover-crop soils. This was replaced by vermiculite, vermiculite mixed layers, and illite. Also, the smectite rich mixed-layer of the conventional granodiorite sample (BG) was apparently replaced by a non-expandable illite–vermiculite under plant cover (CG). In Barré et al. [29], it was reported that a reversible behavior of the illite from the vermiculite phase was attributed to the plant-mediated potassium dynamic, because the process of potassium increasing in grassed C horizons would displace the center of gravity of clay assemblages from ≃1.500 to 1.000 nm. Thus, the mixed layers in general, and those richer in smectite in particular, probably embody complex reaction intermediaries [48]; however, these might play a close role in the stabilisation of the non-hydrolysable carbon sub-fractions, as revealed in the PCNs. The relationship between soil smectite-rich minerals and humic substances has been proved [15,49]. Some studies [50] on fresh marine and poldered materials indicate that the presence of grasses, as in temporary cover-crop soils, could cause the collapse of swelling layers due to the presence of organic matter, an effect that might be reinforced by the plant-mediated potassium enrichment of these soils. This makes it difficult to keep the interlayers open to polar molecules and, subsequently, inhibits their response to glycol.

Although there is a direct, non-spurious, causal relationship between the non-hydrolysable phases and the mixed layers, especially those rich in smectite, it is difficult to establish the mechanism of such interaction. Our results suggest that the interaction with phyllosilicates, rather than the recalcitrance traditionally attributed to some organic molecules [13], prevents the acidic digestion of the <53 μm SOC fraction. The intercalation of SOC prevents the swelling of the interlayers; consequently, the interaction with phyllosilicates would be done at the level of <50 μm clay microstructures, clusters and domains. In this case, it would be a physical occlusion mechanism; but, as Virto et al. [14] stated, it is likely most of the SOC in <50 μm silt-size aggregates is directly bound to clay particles.

The direct relationship established between the hydrolysable fractions of SOC and calcite is remarkable and could be linked to the saturation of calcium in the soil solution, a factor favouring the adsorption of organic carbon molecules on phyllosilicate surfaces [12,16]. On the contrary, particulate SOC (>53 μm) tends to be negatively correlated with MLs. Moreover, iPOM was positively correlated with non-phyllosilicate phases (Q and FdCaNa) and with the less-swelling ML (V/Sm), which might be explained by the lower surface reactivity of these mineral regarding smectite phases. Similarly, the light fraction of SOC (LF) also establishes a direct relationship with primary clay particles, such as vermiculite and chlorite. In general, the phases in which the particulate material showed a positive correlation have less SSA than the minerals correlated with the chemically or bio-chemically protected fractions. This agreed with the high negative correlation that can be obtained between vermiculite phases (MLV plus V) and SSA (r = −0.804, *p* = 0.016), that is also significant when controlling SOC (partial r = −0.824, *p* = 0.023). This could show again the importance of surface processes in SOC stabilization for colloidal or molecular SOC forms at the expense of the particulate carbon.

## 5. Conclusions

Our results suggest that the implementation of spontaneous cover crops in olive orchards is a technically and economically strategic way to increase the levels of SOM and SOC in olive orchards. Our results also provide evidence of the causal interactions that might question some widely accepted SOC stabilization mechanisms. This is the case for the direct correlation between biochemically protected SOC fraction and mixed layers, which could be more associated with mineral surfaces than to the traditional concept of

biochemical recalcitrance. Indeed, causal relationships between functional pools and clay minerals would more likely be given at the microscale (hydrolysable, non-hydrolysable SOC), because no significant correlations were found between the macroscopic pools (NPP, PPP, biochemically and chemically protected) and the clay assemblages, but rather with the organic subfractions (cPOM, LF, NH, H, iPOM, μNH, μH), which composed the macroscopic pools. Based on an analysis with such a small and un-replicated number of samples, it is not trivial to establish that in conventionally managed olive groves with bare soils there is more biochemically protected SOC by effects related to the intimate soil mineralogy. However, these conclusions will require a broader experimental approach.

**Supplementary Materials:** The following supporting information can be downloaded at: https://www.mdpi.com/article/10.3390/min13010060/s1, Table S1: Pearson's correlation coefficients between SOC functional fractions (%), soil properties of the fine earth (>2 mm) and mineral phases in the <2 μm fraction (%). Table S2: Pearson's correlation coefficients between SOC functional fractions (%) and some properties of the <2 μm fraction (%). Table S3: Pearson's correlation coefficients between mineral phases in <2 μm fraction (%) and some properties of the fine earth and <2 μm fraction (%). Table S4: Pearson's correlation coefficients between properties of the fine earth and the <2 μm fraction (%).

**Author Contributions:** Conceptualization, J.C. and R.G.-R.; Formal analysis, J.C., R.G.-R. and J.M.M.-G.; Funding acquisition, R.G.-R.; Investigation, J.C., M.T.-C., J.L.V.-V. and J.M.M.-G.; Methodology, J.C., R.G.-R. and J.M.M.-G.; Writing—original draft, J.C. All authors have read and agreed to the published version of the manuscript.

**Funding:** This research was funded by the PRIMA-H2020 project SUSTAINOLIVE, grant number no1811, and was also possible thanks to the project 1261443 financed by the Programa Operativo FEDER Andalucía 2014–2020.

**Conflicts of Interest:** The authors declare no conflict of interest.

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
