# Peer review of "Role of Clay Mineralogy in the Stabilization of Soil Organic Carbon in Olive Groves under Contrasted Soil Management"

_minerals, doi:10.3390/min13010060_

Round 1

Reviewer 1 Report

As the authors point out, cropland soils are key systems to global carbon budgets due to their high carbon sequestration potential. And it is widely accepted that clay minerals are one of the soil components that have a strong effect on the stabilization of the soil organic carbon, owing to its surface interactions with organic molecules. However, the identification of the direct effects of clays on soil organic carbon stabilization is complicated, due primarily to the difficulty of accurately characterizing the mineralogy of the soil clay minerals. These authors found evidence that both the flocculation of clays and the adsorption of organic carbon depend upon the same electrokinetic and thermodynamical processes developed on mineral surfaces, which were strongly conditioned by clay mineralogy. Therefore, a comprehensive understanding of soil organic carbon stability should consider both the physical and chemical protection mechanisms, which define the different functional pools.

For their research, the authors selected Olive groves in two sites with contrasted soil parent material. The values of the physical and chemical properties indicate that studied soils that are poorly suited for agriculture, with high proportion of coarse fragment, especially in soils over the acidic granodiorite parent materials. As expected, properties of soils under the contrasted parent materials widely differed. Soil pH, equivalent CaCO3, cation exchange capacity and exchangeable potassium were higher in soils under calcareous parent materials.

Following extensive laboratory studies, the values of the physical and chemical properties indicated that soils with high proportion of coarse fragments, especially in soils formed over the granodiorite parent materials are poorly suited for agriculture. Granodiorite soil samples were richer in quartz than calcareous samples, while phyllosilicates tended to predominate in the in the calcareous samples. In the calcareous sites, the carbonates varied according to both the parent material and the soil management, being larger in the marls than in limestone colluvium, and larger in the bare soils than in the soils with cover crops.

Overall, the authors have done an excellent job of defining the soil issues and then proceeding with very detailed and well described laboratory experiments. The figures and the references are excellent. I do have a couple of suggestions to offer that I think will improve the readability of the text:

1.     There are a number of grammar and syntax corrections needed throughout the text. For example, Line 17 should read “complicated, mainly…” Line 20 should begin “The total mineralogy…” Line 42 should read “Despite the fact that cropland…” Line 44 should read “This might account up to 20% per year of the fossil fuel…” And more throughout the text need correction.

2.     I suggest the XRD peaks be identified by their angstrom values rather than nanometer values. This will make them much easier to understand by many mineralogists.

Author Response

Changes introduced in the review process have been marked in red in the manuscript.

Reviewer comment #1. “There are a number of grammar and syntax corrections needed throughout the text. For example, Line 17 should read “complicated, mainly…” Line 20 should begin “The total mineralogy…” Line 42 should read “Despite the fact that cropland…” Line 44 should read “This might account up to 20% per year of the fossil fuel…” And more throughout the text need correction.”

Answer of the authors to comment #1. The text has been revised and these and other grammatical inaccuracies has been corrected.

Reviewer comment #2. “I suggest the XRD peaks be identified by their angstrom values rather than nanometer values. This will make them much easier to understand by many mineralogists.”

Answer of the authors to comment #2. We thank the reviewer his interesting remark, but we have followed the Instructions for Authors of the journal about the use of SI units.

Reviewer 2 Report

Dear Authors,

I understand that the interpretation of this topic is not easy. Therefore, present your results as simply as possible. It is also necessary to draw clear results from your work to say why your article is significant and beneficial.

For more precise comments, see the attached file.

Regards.

Author Response

Changes introduced in the review process have been marked in red in the manuscript.

Reviewer comment #1 (Introduction). “This chapter is well written. However, since the issue of either permanent or temporary carbon storage in the soil is very complex, quote the opinions of a higher number of authors. Focus, in particular, on the principles of organic carbon stabilization in the soil. See for example this: The soil carbon dilemma: Shall we hoard it or use it? https://doi.org/10.1016/j.soilbio.2005.10.008”

Answer of the authors to comment #1 (Introduction): I thank the reviewer for his comment. Actually, I've read the recommended article carefully and it raises an interesting question: how to balance C storage with fluxes. We have introduced this idea and the reference in the revised manuscript (L. 61)

Reviewer comment #2 (Material and Methods). “Why so shallow?”

Answer of the authors to comment #2 (Material and Methods). The choice of this sampling depth was required by the excessive stoniness of some profiles, especially those on granodiorites. In some cases, fine earth amounts were less than 25% w/w, so the profile is reduced to a shallow epipedon of little pedological development over the regolith. Since one the main aims of this study was to characterize the organic matter fraction, we found serious troubles when applying the OC fractionation on samples that practically lacking organic matter at his topsoil (0 - 20 cm).

Reviewer comment #3 (Material and Methods). “The article you cited does not describe the method. It refers to other articles. Therefore, describe the method exactly yourself, or cite the original article that describes the method.”

Answer of the authors to comment #3 (Material and Methods). Sorry, this was a mistake between the reference #22 and #24. This has been check and corrected in the revised manuscript.

Reviewer comment #4 (Material and Methods). “Why was pHKCl or pHCaCl2 not measured? I believe they have a higher value.”

Answer of the authors to comment #4 (Material and Methods). The pH was measured in H2O because it is the standard measure of pH in agricultural soils (Weil & Brady, 2017*, p. 407). However, as a relatively high pool of exchangeable activity was expected in soils on granodiorites, the pH was also measured in 1N KCl (see below). Broadly, the differences between both pH determinations (in absolute value) should be attributed to the exchangeable acidity (Al3+ plus H+ in the CEC): the larger the difference the larger the latter. These data were not included in Table 1 since it was considered that they did not provide any relevant data for the discussion.

Sample

pH

H2O

pH

KCl

DpH

WG

6.18

5.60

-0,58

BG

5.88

4.89

-0,99

CG

6.24

5.50

-0,74

*Weil, R., Brady, I.E., 2017. The Nature and Properties of Soils (15th Edition). Pearson Education Inc., Columbus (OH).

Reviewer comment #5 (Results). “In this case, I would strongly consider combining the Results and Discussion chapters. I think the text would be easier to understand. Focus more on the substance of this article (subsections 3.3 and 3.4).”

Answer of the authors to comment #5 (Results). Due to the number of results exposed and their different nature (clays, organic fraction, etc.), we believe more suitable to keep the separation between chapters 3 (Results) and 4 (Discussion). Nevertheless, the reviewer is right in his comment: the length of the different subsections of chapter 3 are rather unbalanced, mainly subsection 3.2 (Clay mineralogy), whose excessive length prevents an easy understanding of the entire chapter. To avoid this, the last paragraph of subsection 3.2 (Lines 375 to 438 of the original manuscript) has been reallocated and integrated in the Discussion chapter (see below, answer to comment #9), and also other minor changes has been done along chapter 3.

Reviewer comment #6 (Results). “[34,36]”

Answer of the authors to comment #6 (Results). Done (all references has been checked and corrected, see answer to comment #3).

Reviewer comment #7 (Results). “Use the same number of decimal places (sometimes you have one, sometimes two, sometimes three, ...)”.

Answer of the authors to comment #7 (Results). Done.

Reviewer comment #8 (Table 5). “Is it not possible to make a clear statistical evaluation here?” Answer of the authors to comment #8 (Table 5). In this case, we should include averages and standard deviations of type of management (bares soils vs. covered soils) and/or parent material (granodiorites vs. marls vs. limestone colluvia) but, in our opinion, this would make the table less accessible for reading. We propose to maintain table 5.

Reviewer comment #9 (Discussion). “Discussion is poor. It is necessary to base your opinions on professional literature. You don't have to agree with it, but in that case, explain why. In any case, more expert opinions need to be implemented in this chapter.”

Answer of the authors to Answer of the authors to comment #9 (Discussion). We agree. A significant part of this chapter has been reworked: Some paragraphs from “Results” have been taken and integrated here, and other paragraphs of the original manuscript have been removed or rewritten. We believe this improves the chapter, which goes from 95 to 118 lines and from 15 to 20 references.

Reviewer comment #10 (Conclusions). “Be specific in the Conclusion chapter. Select the most important thing from your article.”                                                                                                               

Answer of the authors to comment #10 (Conclusions). We Agree. Conclusions have been summarized and partially rewritten.

Round 2

Reviewer 2 Report

Thank you. I think the revision of the article helped to improve its quality.